# Star Attention: Efficient LLM Inference over Long Sequences

**Shantanu Acharya** [1]  **Fei Jia** [1]  **Boris Ginsburg** [1]

## Abstract

Inference with Transformer-based Large Language Models (LLMs) on long sequences is both costly and slow due to the quadratic complexity of the self-attention mechanism. We introduce Star Attention, a two-phase block-sparse approximation that improves computational efficiency by sharding attention across multiple hosts while minimizing communication overhead. In the first phase, the context is processed using blockwise-local attention across hosts, in parallel. In the second phase, query and response tokens attend to all prior cached tokens through sequence-global attention. Star Attention integrates seamlessly with most Transformer-based LLMs trained with global attention, reducing memory requirements and inference time by up to 11x while preserving 97-100% of accuracy.

## 1. Introduction

Recent Large Language Models (LLMs) can support contexts up to millions of tokens in length (Gemini-Team, 2024; Anthropic, 2024; Meta-AI, 2024; Qwen, 2025), unlocking applications such as repository-level code analysis, multi-document summarization, and large corpus retrieval. However, processing such long sequences with LLMs requires substantial computational and memory resources due to the quadratic complexity of the self-attention mechanism.

The importance of long-context capabilities has driven substantial research into addressing the computational challenges of self-attention. Some approaches focus on reducing the need to fully materialize the attention matrix (Milakov & Gimelshein, 2018), leading to blockwise processing techniques (Dao et al., 2022; Dao, 2024; Liu & Abbeel, 2023) and further optimization through distributed computation across multiple compute units (Liu et al., 2024a). While

---

[1]NVIDIA. Correspondence to: Shantanu Acharya <shantanua@nvidia.com>, Fei Jia <fjia@nvidia.com>.

*Proceedings of the $42^{nd}$ International Conference on Machine Learning*, Vancouver, Canada. PMLR 267, 2025. Copyright 2025 by the author(s).

these methods improve training efficiency, autoregressive decoding during inference still requires the model to attend to every previous token, resulting in high computational costs for long-context sequences. Other approaches attempt to optimize inference by segmenting long inputs into chunks, encoding them separately, and attending to these encoded chunks using the user query (Beltagy et al., 2020; Russak et al., 2024; Liao et al., 2024). However, these methods often require fine-tuning the model or introducing additional components that necessitate further training, limiting their out-of-the-box applicability.

We introduce Star Attention[1], a novel algorithm for efficient LLM long-context inference. This method is based on the observation that LLM inference usually has two stages: (1) prompt encoding, where the model processes input and stores KV vectors in the cache and (2) token generation, where model attends to the KV cache and autoregressively generates new tokens while updating the cache with the new KV vectors. In many long-context tasks, the input consists of a long context followed by a short query and a short answer. The information needed for answering the query is often localized within small parts of the context, meaning context tokens need only attend to nearby tokens, while query tokens need to attend to all prior tokens. Based on this observation, *Star Attention* utilizes a two-phase approach shown in Figure 1:

1. **Context Encoding**: The context is divided into contiguous blocks and distributed across "context" hosts, with each host also receiving a copy of the first block (an "*anchor block*"). Hosts compute self-attention only for their assigned blocks, without communicating with each other, reducing attention complexity from quadratic to linear with respect to context length. This distributed processing is similar to Ring Attention (Liu et al., 2024a) but without the "ring" communication during context encoding (Figure 1a).

2. **Query Encoding and Token Generation**: The query is replicated across all hosts where it initially attends to the KV cache on each host. Global attention is then computed by aggregating the results at a designated "query" host by efficiently communicating a single vec-

---

[1]Code: https://github.com/NVIDIA/Star-Attention

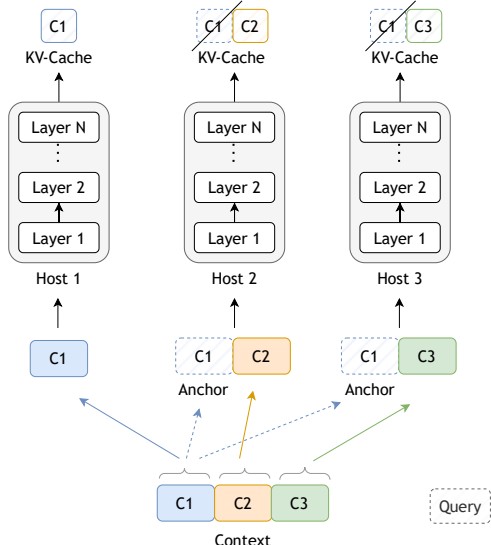

(a) **Phase 1**: Local Context Encoding with Anchor Blocks.

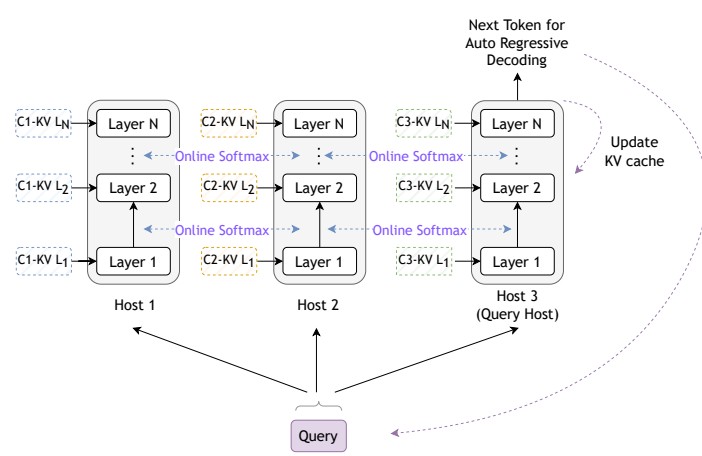

(b) **Phase 2**: Query Encoding and Output Generation with Global Attention.

*Figure 1.* Star Attention inference flow across two phases. (a) *Context Encoding*: The input context is partitioned into blocks and distributed across hosts, where each block (except the first) is prefixed with the anchor block ($c_1$). Each host processes its assigned block and stores the non-anchor portion of the KV cache. (b) *Query Encoding and Token Generation*: The query is broadcast to all hosts, which compute local attention using cached KVs. A designated "query" host then aggregates softmax normalization statistics to compute global attention and generates the next token.

tor and scalar per token from each context host. Only the query host updates its KV cache during this stage (Figure 1b).

Star Attention enables the context length to scale linearly with the number of hosts by distributing the context processing across multiple hosts. Star Attention is compatible with most Transformer-based LLMs trained with global attention, operating seamlessly out-of-the-box without additional model fine-tuning. Furthermore, we combine Star Attention with Flash Attention, allowing for additional speedup enhancements. We evaluate Star Attention for Llama3.1-8B and Llama3.1-70B (Meta-AI, 2024) on several long-context benchmarks. Star Attention achieves up to 11 times faster inference while maintaining 97-100% of the baseline accuracy.

## 2. Star Attention Algorithm

Star Attention operates in two phases: (1) *Context Encoding*, where the long context is divided into contiguous blocks and is processed with local blockwise attention, and (2) *Query Encoding and Token Generation*, where the query is processed, and answer tokens are generated using global attention. Below, we detail each phase of the algorithm.

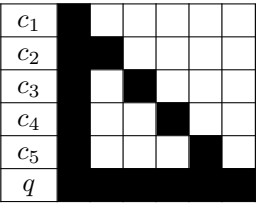

*Figure 2.* Block sparsity pattern in Star Attention for a sequence partitioned into 5 context blocks $c_i$ and a query block $q$. Each context block attends only to itself and the "anchor block" whereas the query attends to the entire input.

### 2.1. Phase 1: Context Encoding

Given an input sequence comprising a context $c$ followed by a query $q$, the context $c$ is divided into $n$ contiguous blocks: $c = [c_1, c_2, \ldots, c_n]$, where each block $c_i$ contains $b$ tokens. We introduce an *anchor block* mechanism, in which, each block—except the first—is prefixed with the first block $c_1$ of the sequence, referred to as the anchor block. This concatenation forms an augmented context $c'$:

$$c' = [c_1, (c_1\ c_2), (c_1\ c_3), \ldots, (c_1\ c_n)]$$

where each augmented block $c'_i$ contains $2b$ tokens: $b$ tokens from the anchor block $c_1$ followed by $b$ tokens from the current block $c_i$ (Figure 2). The positional indices of $c_1$

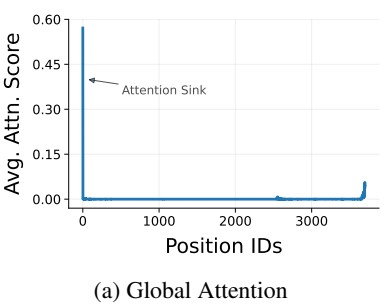

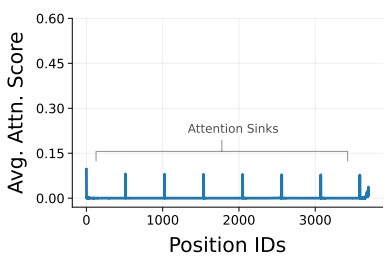

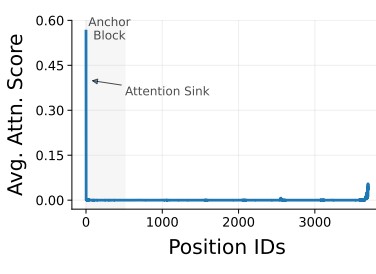

| (a) Global Attention | (b) Blockwise Encoding | (c) Blockwise Encoding w/ Anchor Block |

*Figure 3.* Attention distribution across the sequence during context encoding under different strategies in Phase 1. (a) Global attention exhibits a single attention sink at the sequence start. (b) Without anchor blocks, blockwise context encoding creates multiple attention sinks at the start of each block. (c) With anchor blocks, attention sinks shift to anchor tokens, yielding a distribution that closely approximates global attention. The sequence is 4K tokens long and partitioned into 512-token chunks.

are preserved, ensuring that its tokens retain their original position indices $[0, 1, \ldots, b-1]$. The augmented blocks are distributed across compute hosts, where each host computes attention over the $2b$ tokens from its assigned block $c_i'$ and generates the corresponding key-value (KV) vectors. While KVs for the anchor block $c_1$ are discarded, the KVs for the current block $c_i$ are retained in the cache.

We observe that, without anchor blocks—i.e., applying blockwise attention only to the original context $c$—the model fails to generate correct outputs. We conjecture this failure is due to the incorrect approximation to the attention patterns observed during phase 2 (Figure 3b), where multiple attention spikes, known as attention sinks (Xiao et al., 2024b), are distributed across the sequence. These spikes occur because each block is processed independently, creating an attention sink at the start of each block. As a result, the model struggles to effectively focus on relevant parts of the context. To address this issue, we prefix the blocks with the anchor block $c_1$, shifting the attention sinks to the anchor tokens. By discarding the KVs of the anchor tokens the intermediate attention sinks are removed ensuring the attention distribution of block-local attention (Figure 3c) closely approximates global attention (Figure 3a) while maintaining the computational efficiency of blockwise processing.

### 2.2. Phase 2: Query Encoding and Token Generation

In phase 2, global attention is employed to encode the query and generate output tokens by using a distributed softmax algorithm that eliminates the need to transfer KV cache between hosts (Figure 1b). A designated query-host $h_q$ coordinates this computation. The query is broadcast to all hosts and transformed into the sequence $Q \in \mathbb{R}^{l_q \times d}$, where $l_q$ is the query length, and $d$ is the attention head dimension. Each host $h$ computes the local attention output $A_h$ for the query $Q$ using its local key-value pairs $K_h, V_h \in \mathbb{R}^{l_k \times d}$, where $l_k$ is the sequence length of the KV cache. The local

attention is computed as:

$$A_h = \left( \frac{\exp\left(\frac{QK_h^\top}{\sqrt{d}}\right)}{\sum_{k=1}^{l_k} \exp\left(\frac{QK_{h,k}^\top}{\sqrt{d}}\right)} \right) V_h \qquad (1)$$

In addition to $A_h$, each host also stores the sum of the exponents $s_h$ from the the local softmax operation (the denominator from Equation 1):

$$s_h = \sum_{k=1}^{l_k} \exp\left( \frac{QK_{h,k}^\top}{\sqrt{d}} \right) \qquad (2)$$

The query-host $h_q$ gathers the local attention $A_h$ and the sums of exponents $s_h$ from all hosts:

$$A = [A_1, A_2, \ldots, A_H]$$

$$s = [s_1, s_2, \ldots, s_H]$$

The global softmax denominator, $s_{\text{global}}$, is then computed as the sum of all local exponents:

$$s_{\text{global}} = \sum_{h=1}^{H} s_h \qquad (3)$$

The query-host uses $s_{\text{global}}$ to aggregate the local attentions to compute the global attention:

$$A_{\text{global}} = \sum_{h=1}^{H} \frac{s_h}{s_{\text{global}}} A_h \qquad (4)$$

This method ensures that the global attention scores are normalized correctly across all hosts. It requires the communication of only a single scalar $s_h$ (the local sum of exponents) and a vector $A_h$ (the local attention) per token.

The above formulations provide a simplified conceptual overview. In practice, for efficient inference, we use Flash Attention (Dao, 2024) to attend to the KV cache on each

*Table 1.* Accuracy and relative inference speedup of Star Attention compared to Ring Attention on RULER across sequence lengths from 16K to 128K. Accuracy is reported as the absolute difference from Ring Attention and speedup reflects relative improvements in inference efficiency. Star Attention significantly accelerates inference with minimal accuracy loss.

| Model | Seq. Len. (K) | Block Size (K) | Ring-Attn Acc.(%) | Star-Attn | |
|---|---|---|---|---|---|
| | | | | $\Delta$ Acc. | $\Delta$ Speedup |
| Llama-3.1-8B-Instruct (Meta-AI, 2024) | 16 | 4 | 92.22 | -0.94% | 1.1x |
| | 32 | 8 | 87.53 | +1.17% | 1.2x |
| | 64 | 16 | 84.79 | -1.42% | 1.8x |
| | 128 | 32 | 76.31 | -1.90% | 2.7x |
| Llama-3.1-70B-Instruct (Meta-AI, 2024) | 16 | 4 | 95.09 | -2.71% | 1.7x |
| | 32 | 8 | 94.61 | -2.55% | 2.0x |
| | 64 | 16 | 88.54 | -1.44% | 4.7x |

host and apply the *log-sum-exp* trick from online softmax (Milakov & Gimelshein, 2018) to ensure numerical stability during global attention aggregation.

**Output generation and cache update.** After computing the global attention output, the query-host $h_q$ generates the next token and its KV cache is updated with the key and value vectors of the new token. This process is repeated for each generated token.

This two-phase mechanism—local context encoding with anchor blocks in Phase 1 followed by global query encoding with token generation in Phase 2—gives significant improvements in inference speed, while keeping the accuracy close to the global attention.

## 3. Experiments

We empirically evaluate Star Attention using several Llama-based models across multiple long-context benchmarks with sequence lengths ranging from 16K to 1M tokens, assessing both its accuracy and inference speedup relative to established baselines. We also investigate the accuracy-speed trade-offs as a function of block size and provide a granular breakdown of Star Attention's effectiveness across different domains. Our results demonstrate that Star Attention consistently achieves near-parity with full global attention in accuracy while delivering substantial speedups, especially on large models and long-context tasks.

### 3.1. Setup

**Models.** We conduct experiments using both the base and instruct variants of Llama-3.1 8B which support context lengths up to 128K tokens (Meta-AI, 2024). To evaluate scalability beyond this range, we use gradientai-Llama-3-8B-Instruct-262K and gradientai-Llama-3-8B-Instruct-1048K that extend Llama-3-8B's context to 256K and 1M tokens respectively (Gradient.ai, 2024). We further assess

the impact of model scale using Llama-3.1-70B-Instruct. Across all configurations, Star Attention demonstrates increasing speedup benefits with larger models and longer sequences.

**Baseline.** We compare Star Attention against three strong baselines: (i) Ring Attention (Liu et al., 2024a), a distributed attention mechanism that computes global block-wise attention by circulating each host's KV cache in a ring pattern across all the hosts; (ii) StreamingLLM (Xiao et al., 2024b), a sparse attention method that combines global sink tokens with sliding window attention. We use a configuration having 1000 global sink tokens along with a sliding window of 8000 tokens; and (iii) MInference (Jiang et al., 2024), which utilizes three distinct sparse attention patterns, dynamically selecting the optimal pattern per head in an offline search setting. Among these, only Ring Attention is a distributed algorithm designed to scale inference across multiple GPUs. Since Star Attention also targets distributed efficiency, we report speedup metrics relative to Ring Attention, while accuracy comparisons are provided for all three baselines.

**Configuration.** We implement Star Attention in both HuggingFace Transformers library (Wolf et al., 2020) and NVIDIA's TRT-LLM framework (NVIDIA, 2023). All experiments are conducted on NVIDIA A100 GPUs with bfloat16 precision. Optimization techniques such as Flash Attention are applied uniformly across Star and Ring Attention implementations to ensure a fair comparison. Reported results are based on the HuggingFace implementation, with similar relative trends observed across TRT-LLM. Additional details regarding our experimental setup can be found in Appendix B.

**Evaluation Benchmarks.** We evaluate our method on three benchmarks, each testing unique aspects of long context understanding: (i) RULER (Hsieh et al., 2024): a synthetic benchmark with 13 tasks categorized into 4 domains: Needle-in-a-Haystack (Retrieval), Multi-Hop Tracing, Ag-

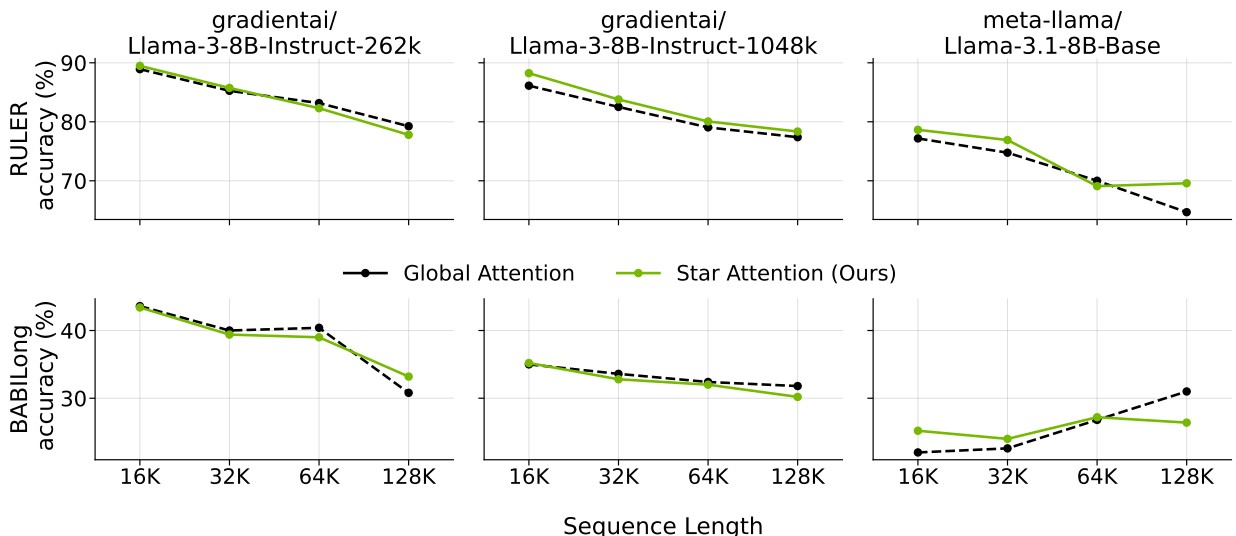

*Figure 4.* Accuracy comparison of Star Attention and Global Attention on RULER and BABILong from 16K to 128K sequence lengths using various models. All runs use a block and anchor block size set to one-quarter of the total sequence length. Star Attention maintains 97-100% of the accuracy of global attention, and in some cases, even outperform it.

gregation, and Question Answering. (ii) BABILong (Kuratov et al., 2024): a benchmark of 5 tasks requiring reasoning over multiple supporting facts encoded in the context to generate accurate answers. (iii) InfiniteBench (Zhang et al., 2024): a diverse collection of 10 real-world and synthetic tasks spanning summarization, multilingual QA, code debugging, and retrieval. Further details on the benchmarks and specific tasks can be found in Appendix C.

### 3.2. Results

Table 1 presents the accuracy and relative speedup of Star Attention compared to Ring Attention (representing full global attention in a distributed setting) on RULER, across sequence lengths from 16K to 128K tokens. In each setting, the context and the anchor block size are set to one-quarter of the total sequence length. Star Attention maintains high accuracy, typically within 0-3% of global attention, while delivering significant speedups, ranging from 1.1× to 4.7×, depending on the model size and sequence length. The speedup becomes more pronounced for larger models. For instance, the Llama-3.1-70B-Instruct model exhibits a 4.7× acceleration at 64K tokens with minimal accuracy drop. This highlights Star Attention's suitability for high-throughput inference and its ability to preserve model's accuracy even with a significantly reduced context window.

To evaluate generalization beyond RULER, we benchmark Star Attention on BABILong using Llama-3.1-8B-Base, gradientai-Llama-3-8B-Instruct-262K, and gradientai-Llama-3-8B-Instruct-1048K. As shown in Figure 4, Star Attention consistently achieves near-parity with full attention across all tasks up to 128K sequence length, with an accuracy drop typically below 3%. However, we observe anomalies for the Llama-3.1-8B base model on BABILong, likely due to format-specific generation requirements that challenge non-instruction-tuned models, particularly at longer sequence lengths.

### 3.3. Comparison with Other Sparse Attention Methods

While our primary comparison focuses on Ring Attention (Liu et al., 2024a) due to its distributed design, we also evaluate Star Attention against two strong non-distributed sparse attention baselines: StreamingLLM (Xiao et al., 2024b) and MInference (Jiang et al., 2024). These methods represent alternative strategies for long-context efficiency under constrained compute budgets and provide complementary perspectives on accuracy trade-offs.

Table 2 reports accuracy on RULER using Llama-3.1-8B-Instruct across sequence lengths from 16K to 128K tokens. Star Attention outperforms both the methods, with the performance gap widening at longer context lengths. Notably, Star Attention maintains accuracy closest to the baseline (full attention) across all settings, demonstrating its robustness in extended-context reasoning.

To assess generalization beyond synthetic tasks, we further evaluate all methods on InfiniteBench. As shown in Table 3, Star Attention achieves the highest average accuracy across 10 diverse tasks spanning summarization, multilingual QA, code debugging, and retrieval. It excels especially in retrieval-heavy tasks such as *PassKey*, *NumRetr*, and *KVRetr* while also delivering competitive results across other categories. These findings highlight Star Atten-

*Table 2.* Accuracy comparison of different methods on RULER from 16K to 128K sequence length using Llama-3.1-8B-Instruct. Star Attention performs closest to Full Attention and outperforms others at longer sequences.

| Methods | 16K | 32K | 64K | 128K | Average |
|---|---|---|---|---|---|
| Full Attn. | 92.22 | 87.53 | 84.79 | 76.31 | 85.21 |
| StreamingLLM | 74.76 | 48.56 | 26.2 | 30.77 | 45.07 |
| MInference | **93.27** | 86.54 | **84.86** | 58.17 | 80.71 |
| **Star Attention** | 91.27 | **88.70** | 83.37 | **74.41** | **84.44** |

*Table 3.* Accuracy comparison of different methods on InfiniteBench using Llama-3.1-8B-Instruct. Star Attention performs closest to Full Attention and outperforms others across all the diverse tasks.

| Methods | En. Sum | En. QA | En. MC | En. Dia | Zh. QA | Code. Debug | Math. Find | Retr. PassKey | Retr. Num | Retr. KV | Avg. |
|---|---|---|---|---|---|---|---|---|---|---|---|
| Full Attn. | 31.91 | 25.92 | 69.43 | 21.5 | 31.95 | 16.75 | 24.29 | 99.15 | 99.66 | 60 | 48.06 |
| StreamingLLM | 30.15 | 10.15 | 41.05 | 8.5 | 22.38 | 8.63 | 17.71 | 2.71 | 5.93 | 0 | 14.72 |
| MInference | 31.04 | 22 | 63.76 | 14.5 | 28.7 | 5.33 | **27.43** | 56.78 | 77.12 | 14 | 34.07 |
| **Star Attention** | **31.85** | **25.92** | **69** | **22** | **30.37** | **24.37** | 26.29 | **93.22** | **96.27** | **45.8** | **46.51** |

tion's ability to generalize beyond synthetic benchmarks and handle real-world, instruction-heavy tasks with long-range dependencies.

### 3.4. Trade-off between accuracy and speed

Figure 5a illustrates the effect of varying block size during context encoding, with the sequence length fixed at 128K tokens. Larger block sizes lead to improved accuracy, highlighting the benefits of increased receptive fields for long-context comprehension.

Empirically, setting the block size to approximately one-quarter of the total sequence length strikes an effective trade-off between accuracy and speed. For sequence lengths exceeding 128K, we fix the block size at 32K tokens to prioritize inference speed. As shown in Figure 6, this configuration allows Star Attention to achieve substantial speedups over Ring Attention while incurring only modest accuracy degradation. For instance, on the RULER benchmark with Llama-3-8B-Instruct-1048K, Star Attention achieves up to $11\times$ speedup while retaining accuracy comparable to Ring Attention. At 1M tokens, the speedup increases to $16.9\times$ with an accuracy drop of just 5.32%.

These findings demonstrate that Star Attention offers flexible control over the accuracy-efficiency trade-off. Larger block sizes allow performance to approach that of global attention, while smaller blocks enable higher throughput for latency-sensitive applications. The appropriate configuration can thus be tuned based on available resources and task requirements. Additional experimental details are provided

in Appendix B.

### 3.5. In-Depth Analysis on RULER Task Categories

To better understand the strengths and limitations of Star Attention, we analyze its performance across different task categories within the RULER benchmark. RULER comprises five categories: Single-NIAH, Multi-NIAH, Multi-Hop Tracing, Aggregation, and Question Answering (QA). Figure 7 reports category-wise accuracy using the Llama-3.1-8B-Instruct model at a sequence length of 32K and a block size of 8K. We observe consistent trends across all sequence lengths, as detailed in Appendix D.

Star Attention performs comparably to global attention in the Single-NIAH, Multi-NIAH, and QA categories. These tasks typically involve localized retrieval or reasoning, where attention primarily operates within or near a single context block. In contrast, Multi-Hop Tracing presents a greater challenge. It requires propagating information across multiple hops within the sequence, demanding effective inter-block communication. Since Star Attention restricts KV-cache access to the local block during context encoding, the model lacks a mechanism for long-range token-to-token aggregation in this phase. Consequently, performance degrades relative to global attention.

Interestingly, Star Attention shows substantial gains in Aggregation tasks, especially those involving frequency analysis or summarization over distributed spans. Its chunk-wise encoding facilitates local aggregation within blocks, which is later synthesized during the global query phase. This

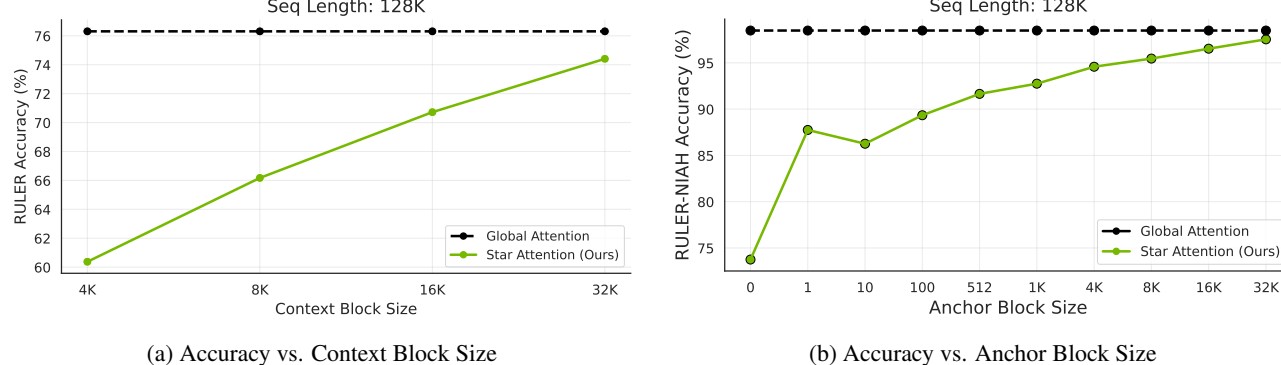

(a) Accuracy vs. Context Block Size

(b) Accuracy vs. Anchor Block Size

*Figure 5.* Impact of context and anchor block sizes on the accuracy of Star Attention at 128K sequence length with Llama-3.1-8B Instruct. (a) Accuracy as a function of context block size, with anchor block size matched to it. (b) Accuracy as a function of anchor block size, with context block size fixed at 32K. Larger block sizes yield consistent accuracy improvements, highlighting the benefit of broader receptive fields for long-context understanding.

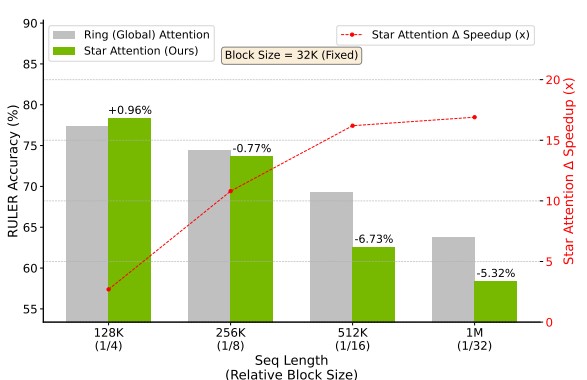

*Figure 6.* Accuracy vs speed trade-off for Star Attention on RULER with Llama3-8B-Instruct-1048K as sequence length increases from 128K to 1M with block size fixed at 32K. Star Attention achieves up to 16.9× speedup with modest accuracy degradation.

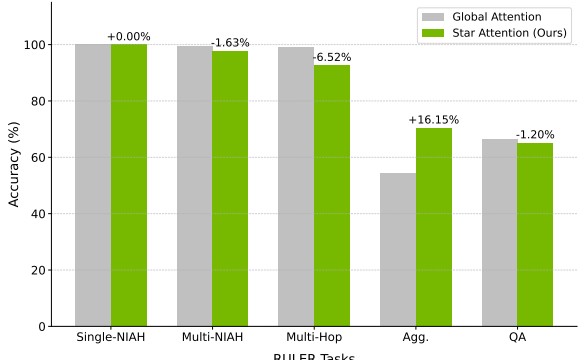

*Figure 7.* Accuracy of Star Attention compared to Global Attention across five RULER task categories using Llama-3.1-8B-Instruct at 32K sequence length with 8K block size. Star Attention matches or improves upon the baseline in most tasks, with significant gains in aggregation.

two-phase process proves advantageous in capturing common patterns without needing full global context at once. This analysis suggests that Star Attention is especially well-suited for retrieval and aggregation tasks, while highlighting opportunities for future work on cross-block communication.

## 4. Ablation Study

The ablation experiments focus on the Needle-in-a-Haystack (NIAH) task, which tests a model's ability to answer queries based on a small, relevant piece of information ("needle") embedded within a large context ("haystack"). To increase the task's complexity, we explore three variations from the RULER benchmark (Hsieh et al., 2024): Single-NIAH, Multi-key NIAH, and Multi-query NIAH.

### 4.1. Position and Content of Anchor Block

In this section, we explore the role of anchor blocks during Phase 1 that enables Star Attention to approximate global attention behavior. As outlined in Section 2.1, anchor blocks are crucial in managing the attention spikes generated at the start of each context block, helping Star Attention approximate global attention (see Table 4 ) Drawing from the hypotheses on sink tokens in Xiao et al. (2024b), we consider two potential explanations for the effectiveness of anchor blocks: (1) the model may develop a bias toward the absolute position of the anchor block, or (2) the semantic content of the anchor block is essential for maintaining performance. To better understand how anchor blocks enable Star Attention to approximate global attention distribution, we test both the hypotheses. We conduct experiments on the Llama-3.1-8B-Instruct model, varying both the position and

*Table 4.* Impact of anchor block position and content on Star Attention accuracy using Llama-3.1-8B-Instruct on RULER-NIAH at 64K and 128K sequence lengths. Each configuration has the anchor block size equal to the context block. The Δ values indicate absolute accuracy degradation relative to global attention. Results show that anchor content is critical, while position IDs haveminor effect. Missing or poorly constructed anchors lead to significant degradation.

| Experiments | RULER-NIAH (%) | | | |
|---|---|---|---|---|
| | 64K | Δ64k | 128k | Δ128k |
| Global attention | 99.50 | - | 98.49 | - |
| No anchor block | 60.11 | -39.59% | 73.75 | -25.12% |
| Content set to first-block, position IDs are: | | | | |
|     randomly sampled from [0, current_block) | 96.79 | -2.72% | 97.16 | -1.35% |
|     same as previous block | 97.35 | -2.16% | 96.80 | -1.71% |
|     **same as first block** | 97.61 | -1.90% | 97.54 | -0.96% |
| Position IDs set to first-block, content is: | | | | |
|     constant token (ex: ' ' or ' the' or '.' ) | 0.00 | -100.00% | 0 | -100.00% |
|     random tokens | 90.55 | -8.99% | 82.63 | -10.15% |
|     shuffled first block tokens | 92.96 | -6.57% | 90.76 | -3.26% |
|     **first block tokens** | 97.61 | -1.90% | 94.94 | -0.96% |
| Previous-block used as anchor | 94.20 | -5.33% | 96.13 | -2.40% |

content of the anchor block. We evaluate two configurations: a block size of 16K for sequences of length 64K, and a block size of 32K for sequences of length 128K, in both the cases, with anchor block size matching the context block size.

**Position of anchor block**: Here, we fix the content of the anchor block to the first context block and vary its position IDs. We test three scenarios : (1) the position IDs are randomly sampled from the range [0, starting position of the current block] (e.g., for a block starting at position 32K, position IDs are sampled from [0, 32K] ); (2) the position IDs are derived from the previous block (e.g., for a block of size 16K starting at position 32K, position IDs are sampled from [16K, 32K] ); (3) the position IDs are fixed to the first block (our proposed approach). As shown in Table 4, varying the position of the anchor block has minimal impact on accuracy.

**Content of anchor block**: We fix the position IDs of the anchor block to that of the first block but vary its content. We explore several configurations (as shown in Table 4): (i) a single repeated token (e.g., ' ', ' the', or '.'); (ii) random tokens; (iii) shuffling the tokens of the first block; and (iv) using the original first block content (the proposed approach). Our results show that the content of the anchor block significantly impacts performance, with the original first block content yielding the best results. This outcome suggests that since global attention is performed during Phase 2, it is important for the local context blocks to attend to anchor blocks whose content reflects what the model would see during global attention.

**Previous block as anchor block**: To examine the roles of both position and content, we experiment with using the previous block as the anchor block. For example, for a block of size 16K starting at position 32K, the anchor block would be the block with position IDs from 16K to 32K. This configuration has lower accuracy comparing to using the first block as the anchor(Table 4).

In summary, we found that while the positional placement of the anchor block is not important , its content is critical for optimal performance.

### 4.2. Size of Anchor block

As discussed in Section 3.4, larger block sizes improve the accuracy of Star Attention. In this section, we analyze the impact of varying anchor block size while maintaining a fixed block size of 32K for a sequence length of 128K. As illustrated in Figure 5b, increasing the anchor block size enhances model accuracy, with the best performance observed when the anchor block size equals the context block size. Although Figure 3b demonstrates that attention spikes predominantly occur in the first few tokens, reducing the number of tokens in the anchor block leads to a substantial drop in performance. This suggests that a larger anchor block is critical for maintaining model accuracy, despite attention spikes being concentrated at the beginning of the sequence. This observation implies that the anchor block's effectiveness is not solely due to its role in managing attention sinks but may involve other underlying factors. These findings remain consistent across both base and instruct models, as well as for all sequence lengths. Further investi-

gation into why the anchor block size must be equivalent to the context block size is left for future work.

## 5. Related Work

To address the computational challenges of long-context inference in LLMs, various techniques have emerged to mitigate memory usage and enhance inference speed.

**Blockwise and Distributed Attention Computation:** Flash Attention (Dao et al., 2022; Dao, 2024) introduces a blockwise GPU-efficient implementation of exact attention, reducing both memory footprint and runtime. Building on this, distributed approaches such as Liu et al. (2024a) and Shyam et al. (2024) partition the computation of self-attention and feed-forward networks across multiple devices, employing sophisticated communication-computation overlap to improve scalability. General distributed strategies (Shoeybi et al., 2019; Huang et al., 2019; Li et al., 2023; Meta-AI, 2021) provide frameworks for dividing the computational load effectively across multiple accelerators. These methods, however, still compute dense global attention, which becomes prohibitively expensive at longer sequence lengths. Star Attention leverages the distributed nature of these approaches but reduces attention complexity through a two-phase block-sparse approximation that avoids computing the full attention matrix.

**Sparse Attention:** Sparse attention methods reduce the quadratic complexity of self-attention through structured or learned sparsity patterns (Zhang et al., 2023; Tang et al., 2024; Child et al., 2019), achieving linear or log-linear scaling in sequence length (Dai et al., 2019; Qin et al., 2024). Beltagy et al. (2020) introduced sliding window attention combined with global tokens, which was adapted by Xiao et al. (2024b) for streaming generation via attention sinks. Jiang et al. (2024) focuses on identifying and leveraging dynamic sparse patterns, particularly to accelerate the pre-filling stage. More recently, Titans (Behrouz et al., 2024) augment LLMs with neural memory modules for long-horizon reasoning. Star Attention's first phase is conceptually similar to streaming methods, but differs by utilizing global attention during decoding—preserving compatibility with pretrained models without retraining.

**Memory Optimization:** Maintaining the KV cache during autoregressive decoding is a major memory bottleneck. KV cache compression (Ge et al., 2024; Munkhdalai et al., 2024; Sun et al., 2024; Liu et al., 2024b; Wu et al., 2024) and low-rank approximation methods (Hu et al., 2022) have been proposed to trade precision for reduced memory. Recent systems explore eviction-based memory management strategies that allow LLMs to operate over virtually infinite contexts (Zhao et al., 2024; Han et al., 2024; Xiao et al., 2024a), often requiring architecture changes or specialized runtime

support. Star Attention is orthogonal to these methods and can be integrated with them to further enhance inference efficiency.

## 6. Conclusion

In this paper, we introduced Star Attention, a novel block-sparse attention mechanism designed to enable efficient inference on long sequences in transformer-based LLMs. The method operates in two phases: (1) context tokens are processed using blockwise-local attention, with the context segmented into blocks where each block is prefixed with an anchor block; and (2) then the query and response tokens attend to all prior cached tokens through sequence-global attention. Star Attention delivers up to 11x speedup over Ring Attention while maintaining 97-100% accuracy, significantly enhancing both memory efficiency and inference speed. Despite these advances, several open questions remain. The role and optimal size of anchor blocks relative to context blocks require further exploration. Additionally, while Star Attention performs effectively with block sizes set to one-quarter of the sequence length, accuracy degrades when using smaller blocks on longer sequences. Future work will focus on refining the anchor block mechanism and improving performance on more complex long-context tasks to enhance the scalability and robustness of Star Attention.

## Acknowledgements

We thank Kefeng Duan, Santiago Akle, Vahid Noroozi, Somshubra Majumdar, Jocelyn Huang, Zhiyuan Jerry Lin and NVIDIA Long Context team for helpful discussion and feedback.

## Impact Statement

This paper presents work whose goal is to advance the field of Machine Learning. There are many potential societal consequences of our work, none which we feel must be specifically highlighted here.

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

# A. Star Attention Pseudo-code

---

**Algorithm 1** Star Attention - Phase 1: Context Encoding

---

**Require:** Context $c$, Block size $b$
1: $L \leftarrow \text{length}(c)$
2: Split $c$ into $n = \lceil L/b \rceil$ blocks, such that $c = [c_1, c_2, \ldots, c_n]$
3: **for** $i = 2$ to $n$ **do**
4:     $c'_i \leftarrow (c_1, c_i)$
5: **end for**
6: **for** each host concurrently **do**
7:     Initialize an empty list $kv$
8: **end for**
9: Distribute augmented blocks $[c'_1, c'_2, \ldots, c'_n]$ across all hosts
10: **for** each host concurrently **do**
11:     **for** each assigned block $c'_i$ **do**
12:         Compute attention over $2b$ tokens in $c'_i$
13:         Generate KV cache for $c'_i$
14:         Discard KV cache for anchor block $c_1$
15:         Append remaining KV cache (for $c_i$) to $kv$
16:     **end for**
17: **end for**

---

**Algorithm 2** Star Attention - Phase 2: Query Encoding and Token Generation

---

**Require:** Query tokens $q$, number of output tokens $n_o$, KV cache $kv_h$ of each host from Phase 1
1: Designate one host as the query-host $h_q$
2: Broadcast query tokens $q$ to all hosts
3: Initialize $input\_tokens \leftarrow q$
4: Initialize $output\_tokens \leftarrow []$
5: **for** $i = 1$ to $n_o$ **do**
6:     **for** each transformer layer **do**
7:         **for** each host $h$ concurrently **do**
8:             Compute query, key, and value vectors $(Q, K, V)$ using $input\_tokens$
9:             **if** $h = h_q$ **then**
10:                 Append the new $K$ and $V$ vectors to $kv_{h_q}$
11:             **end if**
12:             Compute local attention scores $A_h$ for query $Q$ using the local KV cache $kv_h$
13:             Compute local log-sum-exp $s_h$ (logarithm of the softmax denominator)
14:         **end for**
15:         Gather all $A_h$ and $s_h$ from hosts: $s = [s_1, s_2, \ldots, s_H], \quad A = [A_1, A_2, \ldots, A_H]$
16:         Initialize $s_{\text{global}} \leftarrow s_1$, $A_{\text{global}} \leftarrow A_1$
17:         **for** $h = 2$ to $H$ **do**
18:             Update global log-sum-exp $s_{\text{global}}$ using online softmax:
$$s_{\text{global}} \leftarrow s_{\text{global}} + \log\left(1 + \exp(s_h - s_{\text{global}})\right)$$

19:             Update global attention scores:
$$A_{\text{global}} \leftarrow \exp(s_h - s_{\text{global}}) \cdot A_{\text{global}} + \exp(A_h - s_{\text{global}}) \cdot A_h$$

20:         **end for**
21:     **end for**
22:     Generate the next output token and append it to $output\_tokens$
23:     Set $input\_tokens \leftarrow [\text{new output token}]$
24: **end for**
25: **return** $output\_tokens$

---

*Table 5.* Accuracy versus speed trade-off for Star Attention compared to Ring Attention on RULER. The $\Delta$ for star attention shows the absolute accuracy degradation and the relative speedup compared to the baseline. When the block size remains fixed and the sequence length increases, Star Attention achieves exponential speedup over Ring Attention at the cost of slightly more accuracy degradation.

| Model | Seq. Len. (K) | Block Size (K) | Ring-Attn Acc. (%) | Star-Attn | |
|---|---|---|---|---|---|
| | | | | $\Delta$ Acc. | $\Delta$ Speedup |
| Llama3-8B-Instruct, 1048K (Gradient.ai, 2024) | 128 | 32 | 77.39 | +0.96% | 2.7x |
| | 256 | 32 | 74.44 | -0.77% | 10.8x |
| | 512 | 32 | 69.30 | -6.73% | 16.2x |
| | 1024 | 32 | 63.70 | -5.32% | 16.9x |
| Llama-3.1-70B-Instruct, 128K (Meta-AI, 2024) | 64 | 16 | 88.54 | -1.44% | 4.7x |
| | 128 | 16 | 65.29 | -7.47% | 8.7x |

*Table 6.* Time per sample (seconds) for Llama3.1-8B-Instruct model with dense, ring, and star attention, using 8 A100 GPUs. Vanilla autoregressive generation encounters out-of-memory (OOM) at 128K sequence length. It performs best in short context scenarios (i.e. sequences upto 32K tokens) but in long context scenarios, star attention demonstrates significant speedup.

| Seq. Length (K) | Time Per Sample (s) | | |
|---|---|---|---|
| | Vanilla | Ring | Star |
| 16 | 7 | 10 | 9 |
| 32 | 10 | 12 | 10 |
| 64 | 18 | 22 | 12 |
| 128 | OOM | 53 | 20 |

## B. Experiment Details

### B.1. Baseline Comparison

Our implementation utilizes the HuggingFace Transformers library (Wolf et al., 2020), which currently lacks support for multi-node inference. As a result, when performing inference with the Llama-3.1 8B model using standard causal autoregressive generation on sequences exceeding 64K tokens with bfloat16 precision across 8 A100 GPUs, we encounter out-of-memory (OOM) errors. Given these limitations, we adopt Ring Attention as a practical and relevant baseline for evaluating Star Attention's performance on sequences up to 1 million tokens in length.

Table 5 shows speedup obtained by Star Attention over the baseline on sequences over 128K tokens. For such long sequences, we freeze the block size to 32K sequences to optimize for speed. This setting shows upto 16.9x inference speedup with just 5.32% accuracy degradation compared to the baseline. Table 6 presents the time per sample for vanilla autoregressive generation, Ring Attention, and Star Attention across sequence lengths ranging from 16K to 128K. The results indicate that both Ring and Star Attention can process sequences up to 128K tokens on 8 A100 GPUs, whereas vanilla autoregressive inference encounters OOM issues beyond 64K tokens. For sequence lengths below 32K, vanilla inference is faster than the distributed attention mechanisms, primarily due to the GPU communication overhead incurred in the distributed setups. However, in long context scenarios i.e. on sequence lengths exceeding 32K tokens, Star Attention begins to demonstrate clear performance advantages. As demonstrated in Table 5, the speedup achieved by Star Attention increases significantly with longer sequence lengths.

### B.2. Hardware for Inference Speed

We use A100 GPUs to run all our inference speedup experiments. Table 7 describes the number of GPUs and the number of parallel workers used to obtain the inference speed numbers for Ring Attention and Star Attention for each sequence length. In all these experiments, the anchor block size in Star Attention was kept same as the context block size.

*Table 7.* Resources used for the speedup experiments

| Model Size | Seq. Length | # GPUs | # Workers |
|---|---|---|---|
| | 16K - 128K | 8 | 4 |
| 8B | 256K - 512K | 16 | 8 |
| | 1M | 32 | 16 |
| | 16K - 32K | 8 | 4 |
| 70B | 64K | 16 | 4 |
| | 128K | 32 | 8 |

### B.3. Prompt Templates

Prompt template for base models:

```
1  {context}{query}{answer_prefix}
```

Prompt template used for Llama-3 and Llama-3.1 Instruct models:

```
1  <|begin_of_text|><|start_header_id|>system<|end_header_id|>
2
3  You are a helpful assistant.<|eot_id|><|start_header_id|>user<|end_header_id|>
4
5  {context}{query}<|eot_id|><|start_header_id|>assistant<|end_header_id|>
6
7  {answer_prefix}
```

The portion in **blue** is processed during Phase 1 for blockwise context encoding, while the remaining text in **gray** is processed in Phase 2 for query encoding and token generation. The **{context}** and **{query}{answer_prefix}** denote the context and the query portion of the input prompt, respectively. The **{answer_prefix}** is only relevant for the RULER benchmark.

## C. Evaluation Benchmarks

**RULER:** This benchmark comprises 13 tasks covering domains such as Needle-in-a-Haystack (Retrieval), Multi-Hop Tracing, Aggregation, and Question Answering. Each task comprises 500 samples. For the ablations, we choose four Needle-In-A-Haystack (NIAH) tasks where Paul Graham essays serve as the distractor text (haystack): Single 2, Single 3, MultiKey 1, and MultiQuery. In these tasks, a key-value pair is concealed within a long context, and the model must identify the value corresponding to the key based on the provided input query. Table 8 presents the configurations of all the tasks in RULER.

**BABILong:** In BABILong, we choose 5 tasks (shown in Table 9), each containing a 1000 samples. These tasks are generated by simulating a set of characters and objects engaged in various movements and interactions across multiple locations. Each interaction is represented by a factual statement, and the objective is to answer questions based on the facts derived from the current simulation.

**InfiniteBench**: This benchmark comprises 10 real-world and synthetic tasks, each crafted to assess different aspects of language processing and comprehension in extended contexts. Details of each task is shown in 10

*Table 8.* Configuration of RULER tasks

| Category | Task Name | Haystack Type | Keys Type | # | Values Type | # | # Outputs |
|---|---|---|---|---|---|---|---|
| NIAH (Retrieval) | Single 1 | noise | words | 1 | numbers | 1 | 1 |
| | Single 2 | book | words | 1 | numbers | 1 | 1 |
| | Single 3 | book | words | 1 | uuids | 1 | 1 |
| | MultiKey 1 | book | words | 4 | numbers | 1 | 1 |
| | MultiKey 2 | line | words | $\infty$ | numbers | 1 | 1 |
| | MultiKey 3 | kv | uuids | $\infty$ | uuids | 1 | 1 |
| | MultiValue | book | words | 1 | numbers | 4 | 1 |
| | MultiQuery | book | words | 4 | numbers | 1 | 4 |
| Multi-Hop Tracing | Variable Tracking | | | | – | | |
| Aggregation | Common Words Extraction | | | | – | | |
| | Frequent Words Extraction | | | | – | | |
| Question Answering | QA 1 (squad) | | | | – | | |
| | QA 2 (hotpotqa) | | | | – | | |

*Table 9.* Configuration of tasks in BABILong

| Task | Name | # Facts per task |
|---|---|---|
| qa1 | single supporting fact | 2 - 10 |
| qa2 | two supporting facts | 2 - 68 |
| qa3 | three supporting facts | 4 - 32 |
| qa4 | two arg relations | 2 |
| qa5 | three arg relations | 2 - 126 |

*Table 10.* Configuration of tasks in InfiniteBench

| Task name | Context | # samples | Avg. input tokens | Avg. output tokens |
|---|---|---|---|---|
| En.Sum | Fake Book | 103 | 171.5k | 1.1k |
| En.QA | Fake Book | 351 | 192.6k | 4.8 |
| En.MC | Fake Book | 229 | 184.4k | 5.3 |
| En.Dia | Script | 200 | 103.6k | 3.4 |
| Zh.QA | New Book | 175 | 2068.6k | 6.3 |
| Code.Debug | Code Document | 394 | 114.7k | 4.8 |
| Code.Run | Synthetic | 400 | 75.2k | 1.3 |
| Math.Calc | Synthetic | 50 | 43.9k | 43.9k |
| Math.Find | Synthetic | 350 | 87.9k | 1.3 |
| Retrieve.PassKey | Synthetic | 590 | 122.4k | 2.0 |
| Retrieve.Number | Synthetic | 590 | 122.4k | 4.0 |
| Retrieve.KV | Synthetic | 500 | 89.9k | 22.7 |

# D. RULER Analysis

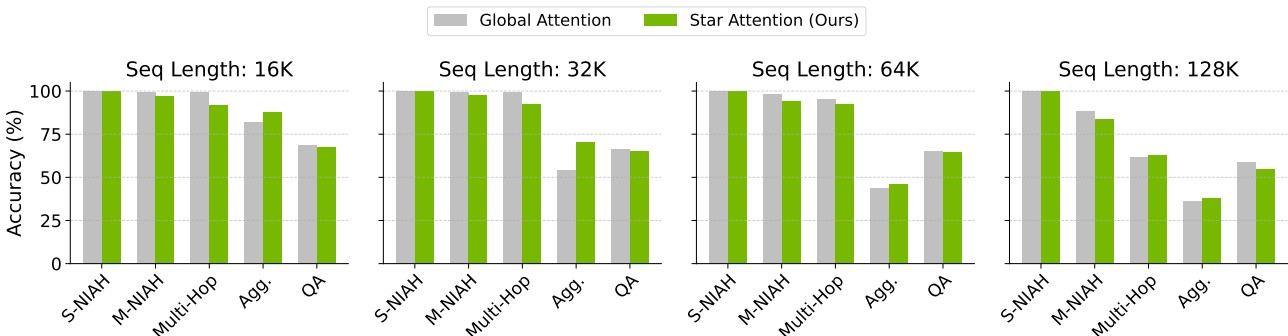

*Figure 8.* Accuracy of Star Attention using Llama-3.1-8B-Instruct on the 5 categories of tasks in RULER on sequence lengths of 16K, 32K, 64K, and 128K. In all experiments, the block size and anchor block size are set to one-quarter of the total sequence length. For the NIAH and QA tasks, Star Attention retains upto 97-100% accuracy of the baseline. The Multi-Hop Tracing task is notably challenging because it requires inter-block communication, which leads to expected performance degradation. Interestingly, Star Attention performs better with sequence lengths of 128k on this task, but this may be due to noise given the suboptimal baseline. In aggregation tasks, Star Attention show significant improvement as distributed local attention helps the model in such summarization tasks.

