# OpenReview forum: "Star Attention: Efficient LLM Inference over Long Sequences"
_ICML.cc/2025/Conference — ICML 2025 poster_

### Official Review · Reviewer_dKGV · 2025-03-06

**Overall Recommendation:** 2

**Summary:**

This paper proposes Star Attention, which improves the LLM inference efficiency by sharding attention across multiple hosts.

**Claims And Evidence:**

Please see **Other Strengths And Weaknesses**.

**Essential References Not Discussed:**

Please see **Other Strengths And Weaknesses**.

**Experimental Designs Or Analyses:**

Please see **Other Strengths And Weaknesses**.

**Methods And Evaluation Criteria:**

Please see **Other Strengths And Weaknesses**.

**Other Comments Or Suggestions:**

Please see **Other Strengths And Weaknesses**.

**Other Strengths And Weaknesses:**

**Strengths**:
1. The paper is easy to follow, with clear writing and presentation.
2. Evaluation results are good.

**Weaknesses**:

1. The main concern I have with this paper is the lack of system performance analysis of proposed method. While the authors present comprehensive algorithm results over various benchamarks, a more detailed performance breakdown at kernel-level would make this method more convincing. Moreover, since sequence parallelism depends on hardware setup, such as GPU accelerator and interconnect (PCIe/NVLink), a more fine-grained analysis for different settings would also provide insights.

2. The literature survey is also not comprehensive.
For instance, there are many other works that focuses on KV cache optimization from architecture/system angeles [1-3].

3. The authors should also further discuss its compatibilty with existing parallelism methods [4-5] and popular LLM serving frameworks [6-7].

4. The title of the paper should emphasize '**distributed LLM inference**' as the proposed method demands multi-GPU/node setup to work.


[1] ALISA: Accelerating Large Language Model Inference via Sparsity-Aware KV Caching, ISCA 2024.

[2] InfiniGen: Efficient Generative Inference of Large Language Models with Dynamic KV Cache Management, OSDI 2024.

[3] FlashInfer: Efficient and Customizable Attention Engine for LLM Inference Serving, Arxiv 2025.

[4] Megatron-LM: Training Multi-Billion Parameter Language Models Using Model Parallelism, Arxiv, 2019.

[5] GPipe: Efficient Training of Giant Neural Networks using Pipeline Parallelism, NeurIPS 2019.

[6] Efficient Memory Management for Large Language Model Serving with PagedAttention, SOSP 2023.

[7] SGLang: Efficient Execution of Structured Language Model Programs, NeurIPS 2024.

**Questions For Authors:**

Please see **Other Strengths And Weaknesses**.

**Relation To Broader Scientific Literature:**

Please see **Other Strengths And Weaknesses**.

**Theoretical Claims:**

Not applied here.

---

> ### Author Rebuttal · Authors · 2025-03-31
>
> We thank Reviewer dKGV for their detailed and insightful feedback. Below, we respond to the concerns regarding system performance analysis, literature coverage, compatibility, and presentation.
>
> ### **1. System Performance Analysis:**
> - While we agree that kernel-level profiling and analysis across hardware configurations (e.g., PCIe vs. NVLink interconnects) can provide valuable insights, our primary focus in this work is to introduce and validate the Star Attention algorithm from an algorithmic and end-to-end performance standpoint. We view detailed kernel-level profiling and hardware-specific optimization as an exciting avenue for future work that could further enhance deployment efficiency.
> - All experiments were conducted under consistent hardware and software environments for Star Attention and the baselines, ensuring that the reported relative speedups (e.g., up to 11x on Llama-3.1-8B-Instruct for long contexts) fairly capture the algorithmic advantage.
> - Notably, Star Attention’s phase 1 avoids inter-host communication entirely, unlike Ring Attention. Even assuming an infinitely fast interconnect, global attention still incurs computation proportional to the total number of tokens per host, while Star Attention reduces this via localized blockwise attention in phase 1.
>
> ### **2. Literature Survey (KV Cache Optimization):**
> - Thank you for the helpful references [1–3]. We will revise the related work section to include and discuss these references.
> - Based on the reviewer's feedback, we conducted new comparisons between Star Attention and other sparse KV cache methods such as StreamingLLM and MInference, on long-context inference tasks. *Star Attention outperforms these alternatives in both accuracy and scalability*, as shown below:
>
> **Table B: Accuracy on RULER (Llama-3.1-8B-Instruct)**
> | Methods            |  16K  |  32K  |  64K  |  128K | Average |
> | :----------------- | :---: | :---: | :---: | :---: | :-----: |
> | *Full Attn. (Baseline)* | *92.22* | *87.53* | *84.79* | *76.31* | *85.21* |
> | StreamingLLM       | 74.76 | 48.56 | 26.2  | 30.77 |  45.07  |
> | MInference         | **93.27** | 86.54 | **84.86** | 58.17 |  80.71  |
> | **Star Attention** | 91.27 | **88.70** | 83.37 | **74.41** | **84.44** |
>
> We also evaluated Star Attention on the InfiniteBench benchmark. Given its broad coverage across multilingual and programmatic tasks, we include this for further evidence of Star Attention’s generalization:
>
> **Table C: Accuracy on Infinite Bench (Llama-3.1-8B-Instruct)**
> | Methods            | En. Sum | En. QA | En. MC | En. Dia | Zh. QA | Code. Debug | Math. Find | Retr. PassKey | Retr. Num | Retr. KV |  Avg. |
> | :----------------- | :-----: | :----: | :----: | :-----: | :----: | :---------: | :--------: | :-----------: | :-------: | :------: | :---: |
> | *Full Attn.  (Baseline)* | *31.91* | *25.92* | *69.43* | *21.5* | *31.95* | *16.75* | *24.29* | *99.15* | *99.66* | *60* | *48.06* |
> | StreamingLLM       |  30.15  | 10.15  | 41.05  |   8.5   | 22.38  |    8.63     |   17.71    |     2.71      |   5.93    |    0     | 14.72 |
> | MInference          |  31.04  |   22   | 63.76  |  14.5   |  28.7  |    5.33     | **27.43** |     56.78     |   77.12   |    14    | 34.07 |
> | **Star Attention** | **31.85** | **25.92** | **69** | **22** | **30.37** | **24.37** |   26.29    | **93.22** | **96.27** | **45.8** | **46.51** |
>
> ### **3. Compatibility with Parallelism Methods and Serving Frameworks:**
> - **Model Parallelism [4, 5]:** Star Attention is complementary to tensor and pipeline parallelism. These can be applied within each host, while Star Attention governs how input context is distributed across hosts via sequence parallelism. No assumptions are made about the intra-host setup.
> - **Serving Frameworks [6-7]:** Star Attention is compatible with modern LLM serving frameworks. For instance, memory management techniques like PagedAttention [6] can be applied within each host to efficiently handle the local KV cache. Because Star Attention distributes context blocks across hosts and isolates them during Phase 1, such memory-optimized serving layers can operate independently within each node. Similarly, the two-phase structure of Star Attention aligns well with structured execution frameworks like SGLang [7], and can potentially be integrated into such systems without requiring fundamental changes to their scheduling or runtime semantics.

---

> > ### Comment · Reviewer_dKGV · 2025-04-02
> >
> > Thank you for your response. However, I will maintain my score.

---

### Official Review · Reviewer_EMsM · 2025-03-12

**Overall Recommendation:** 4

**Summary:**

The paper introduces StarAttention, a sparse attention method for encoding long-context by distributing chunks of context over GPUs.  Unlike Ring Attention, Star Attention uses only local (in-chunk) attention for the majority of the context, allowing for a substantial speedup. Each block attends only to itself and an anchor block; the query at the end of the long input then attends over all input chunks, using a lazy softmax accumulation. Star Attention substantially reduces latency at a small performance cost; ablations show that maintaining an anchor block of meaningful context and limiting the number of total chunks is important to maintaining performance.

**Claims And Evidence:**

Yes; however, I think the last line of the abstract could be worded more carefully-- it currently almost seems to imply that memory requirements are *also* reduced by 11x. I think this is a miscommunication and not an overclaim.

**Essential References Not Discussed:**

I think there could be more discussion of streaming methods for long context that evict the middle cache, keeping only the sink + local context-- these are conceptually adjacent, although they do differ from Star Attention in that they fully evict the middle cache. Some notable examples would be [StreamingLLM](https://arxiv.org/abs/2309.17453) (which is already cited), [LM-infinite](https://arxiv.org/abs/2308.16137), and [InfLLM](https://arxiv.org/abs/2402.04617).

[TurboRAG](https://arxiv.org/abs/2410.07590) may also be relevant, although it also differs significantly from this setting-- in particular, I believe they train their model to adapt to sparse attention patterns from stacking reused KV caches from pre-encoded documents.

**Experimental Designs Or Analyses:**

Yes; I think RingAttention is an appropriate comparison point, and using Flash Attention for each is appropriate.

**Methods And Evaluation Criteria:**

Yes; I think evaluating on RULER and BABILong is reasonable, although it would be a bonus to also see results on a less synthetic task.

**Other Comments Or Suggestions:**

(Related to the questions below) I think more understanding of how many blocks can be used before performance breaks down would be helpful. The settings proposed (generally 4 context blocks, with an anchor block of the same size as the context blocks) seems like a reliable setting, but it would be nice to understand how much this method is robust to varying the block size.

**Other Strengths And Weaknesses:**

I think the ablation of what should go in the context block (and whether its positional IDs matter) is interesting and useful! I appreciated the analysis.

**Questions For Authors:**

Q1. Given a fixed context length, how does the performance vary with block size? Given a fixed block size (e.g. 16k tokens), how many blocks can you add before performance severely declines? There is some discussion of this (and it seems that generally less blocks is better), but how sharp is the dropoff? It would be helpful to understand how dramatic this performance dropoff is.

Q2. You state that a larger anchor block size is critical to maintaining accuracy, and so the anchor block should be the same size as the context block. Is there any benefit to having the anchor block *larger* than the context block? (e.g. if I can only process 32k tokens on each single parallel worker, is it best to have 16k anchor + 16k context, or would 24k anchor + 8k context be better?).

**Relation To Broader Scientific Literature:**

I think this is a nice contribution to efficient long context; while there have been a number of prior works that encode context in chunks and then do some kind of aggregation by retrieval, fusion, or overlapping, this is (to my knowledge) the first work to apply the attention sink + local context method to efficiently prefill a long input's KV cache.

**Theoretical Claims:**

N/A

---

> ### Author Rebuttal · Authors · 2025-03-31
>
> We thank the reviewer EMsM for their insightful comments and suggestions. We address each point below:
>
> ### **1. Clarity of Abstract Wording:**
> Thank you for pointing out the ambiguity in the abstract's final sentence. You are correct that the “up to 11x” improvement specifically refers to inference speed and throughput, not memory. While Star Attention does reduce memory usage (due to sharded attention and localized KV caching), the 11x figure quantifies speedup. We will revise the abstract to explicitly state that the 11x gain pertains to speed/throughput, and mention memory reduction separately, to prevent misinterpretation.
>
> ### **2. Evaluation Benchmarks:**
> We agree that results on real-world or less synthetic benchmarks are valuable. In response, we have extended our evaluation to include InfiniteBench, a long-context benchmark with diverse and challenging tasks. Additionally, we also added comparisons to some of the other sparse attention methods as well such as StreamingLLM and MInference. The results are summarized in the table below.
>
> **Table B: Accuracy on Infinite Bench (Llama-3.1-8B-Instruct)**
> | Methods            | En. Sum | En. QA | En. MC | En. Dia | Zh. QA | Code. Debug | Math. Find | Retr. PassKey | Retr. Num | Retr. KV |  Avg. |
> | :----------------- | :-----: | :----: | :----: | :-----: | :----: | :---------: | :--------: | :-----------: | :-------: | :------: | :---: |
> | *Full Attn.  (Baseline)* | *31.91* | *25.92* | *69.43* | *21.5* | *31.95* | *16.75* | *24.29* | *99.15* | *99.66* | *60* | *48.06* |
> | StreamingLLM       |  30.15  | 10.15  | 41.05  |   8.5   | 22.38  |    8.63     |   17.71    |     2.71      |   5.93    |    0     | 14.72 |
> | MInference          |  31.04  |   22   | 63.76  |  14.5   |  28.7  |    5.33     | **27.43** |     56.78     |   77.12   |    14    | 34.07 |
> | **Star Attention** | **31.85** | **25.92** | **69** | **22** | **30.37** | **24.37** |   26.29    | **93.22** | **96.27** | **45.8** | **46.51** |
>
> ### **3. Discussion of Related Work:**
> Thank you for suggesting the additional relevant references. We will amend the related work section in the revised manuscript to include a more detailed discussion and comparison with related methods.
>
> ### **4. Impact of Block Size and Number of Blocks (Q1):**
> - Figure 5 illustrates accuracy as a function of block size, holding the total sequence length fixed. We observe that larger blocks result in better approximation to global attention and higher accuracy.
> - Figure 6 and Table 5 (Appendix C.1) explore scaling behavior with fixed block size (e.g., 32K) while increasing sequence length up to 1M tokens. We find that Star Attention retains upto 90% of full attention accuracy even at 1M tokens, with up to 17x speedup.
>
> This suggests a graceful degradation, not a sharp drop-off. The method remains robust up to at least 1M tokens, especially with larger blocks.
>
> ### **5. Size of Anchor Block relative to the Block Size:**
> Since Phase 1 uses causal attention (as in decoder-only LLMs), anchor blocks cannot extend beyond the preceding context block without violating causality. Setting the anchor size equal to the context block size ensures maximal usable context per block while preserving autoregressive constraints, effectively bringing Star Attention’s receptive field closer to that of full attention.

---

> > ### Comment · Reviewer_EMsM · 2025-04-02
> >
> > Thanks for the detailed response! I read the response and the other reviews, and I will maintain my positive rating.

---

### Official Review · Reviewer_dnhh · 2025-03-12

**Overall Recommendation:** 2

**Summary:**

This paper propose star-attention which combines a streamingllm attention for the prefill stage and a dense attention for the decoding stage. Specifically, the author implement the streamingllm pre-fill with blocks, where the computing are partioned across the query dimension. The sink and local blocks are packed and distributed, this implementaion, compared to ring-attention requires less communication and thus avoid the ring style data transfer. For the decoding stage, the query are distributed and lse and other inner states are transfered back. The experimental results show that it achieve about 95% accuracy while being upto 11x faster than ring-attention.

**Claims And Evidence:**

-

**Essential References Not Discussed:**

-

**Experimental Designs Or Analyses:**

-

**Methods And Evaluation Criteria:**

The strength of this paper:
1. This paper propose a solid sparse attention approach that does pre-fill sparsely while does decoding densely. And experimental results demonstrate the effectiveness of this approach.

2. The proposed method is especially suitable for distributed system where the block-wise strategy can be applied directly for this method.

The limitation of this paper:
1. The proposed star-attention is simply a streamingllm + a dense bottom window. It's very similar to the vanilla streamingllm and exactly the same compared to tri-shape attention. In general, the proposed algorithm does not provide novel knowledge to the community.

2. The author fail to qualify the motivation of dense decoding verse sparse pre-fill. Unlike tri-shape attention, which provides a good reason for why decoding phase requires a complete access of all history tokens.

3. The author fail to include reasonable baselines to compare with. Star-attention as a lossy attention variant should be compared against for example streamingllm, Minference, or more recent flex-prefill, instead of comparing against ring-attention.

4. The author only report the performance on the the synthetic benchmark RULER, which may fail to test all aspects of star-attention. Should also report metrics on more comprehensive testset such as Infinite-bench, scbench etc.

**Other Comments Or Suggestions:**

-

**Other Strengths And Weaknesses:**

-

**Questions For Authors:**

1. What is the motivation of dense query and sparse pre-fill?
2. How well star-attention perform against other sparse attention baselines? (see above.)
3. How well start-attention perform on comprehensive long-context benchmarks? (see above.)

**Relation To Broader Scientific Literature:**

-

**Theoretical Claims:**

-

---

> ### Author Rebuttal · Authors · 2025-03-31
>
> We thank the reviewer dnhh for their feedback and acknowledge their points regarding novelty, motivation, baselines, and benchmarks. We address these points below:
>
> ### **1. Novelty and Relation to Prior Work:**
> - While Star Attention draws inspiration from prior work like StreamingLLM and attention sinks, it introduces a distinct two-phase distributed inference architecture. Unlike StreamingLLM, which struggles with long-context retention and is not inherently distributed, Star Attention processes the context in parallel without inter-host communication in Phase 1 and leverages global attention only during decoding. This results in both accuracy preservation and significant latency reduction.
> - Regarding “tri-shape attention,” we were unable to identify a specific paper with that terminology. If you are referring to a particular method, we would appreciate a citation to better contextualize and compare it in the final version.
>
> ### **2. Motivation for Sparse Prefill vs. Dense Decoding:**
> - In Phase 1 (sparse prefill), attention is localized, as context tokens generally require only local neighborhood interactions. This allows efficient processing via distributed blockwise attention. In contrast, during query encoding and decoding (Phase 2), the tokens must integrate information from the entire context, necessitating dense attention. This design reflects practical needs in long-context tasks and allows us to optimize for both throughput and accuracy.
> - Conversely, during the query processing and response generation phase (decoding phase), the query tokens and subsequent generated tokens often require access to information scattered throughout the entire preceding context to formulate an accurate response. Therefore, employing dense global attention in Phase 2, accessing the full cached KV state from Phase 1, is crucial for preserving the model's understanding and generation capabilities, particularly for tasks requiring synthesis of information from distant parts of the context.
>
> ### **3. Comparison with Sparse Attention Baselines:**
> - In response to the reviewer’s feedback, we extended our baseline comparisons to include StreamingLLM and MInference. Results are shown in Table A.
>
> **Table A: Accuracy on RULER (Llama-3.1-8B-Instruct)**
> | Methods            |  16K  |  32K  |  64K  |  128K | Average |
> | :----------------- | :---: | :---: | :---: | :---: | :-----: |
> | *Full Attn. (Baseline)* | *92.22* | *87.53* | *84.79* | *76.31* | *85.21* |
> | StreamingLLM       | 74.76 | 48.56 | 26.2  | 30.77 |  45.07  |
> | MInference         | **93.27** | 86.54 | **84.86** | 58.17 |  80.71  |
> | **Star Attention** | 91.27 | **88.70** | 83.37 | **74.41** | **84.44** |
>
> - The results show that Star Attention outperforms all the other sparse KV methods. Furthermore, due to the distributed nature of Star Attention, since there is no inter-block communication during phase 1, the latency on 128K sequence length is equivalent to the latency that the model will have when processing a sequence length of 64K (32K block + 32K anchor), since each host will process each block parallely.
> - Unlike methods such as MInference, which may require offline analysis to determine optimal sparsity patterns, Star Attention can be applied directly to most pretrained Transformer models without model-specific tuning or preprocessing.
>
> ### **4. Evaluation on Comprehensive Benchmarks:**
> - In response to the reviewer’s suggestion, we expanded our evaluation to include InfiniteBench—a more comprehensive benchmark suite for long-context performance. As shown in Table B, Star Attention outperforms other sparse baselines across diverse tasks, demonstrating its robustness beyond synthetic datasets like RULER and BABILong.
>
> **Table B: Accuracy on Infinite Bench (Llama-3.1-8B-Instruct)**
> | Methods            | En. Sum | En. QA | En. MC | En. Dia | Zh. QA | Code. Debug | Math. Find | Retr. PassKey | Retr. Num | Retr. KV |  Avg. |
> | :----------------- | :-----: | :----: | :----: | :-----: | :----: | :---------: | :--------: | :-----------: | :-------: | :------: | :---: |
> | *Full Attn.  (Baseline)* | *31.91* | *25.92* | *69.43* | *21.5* | *31.95* | *16.75* | *24.29* | *99.15* | *99.66* | *60* | *48.06* |
> | StreamingLLM       |  30.15  | 10.15  | 41.05  |   8.5   | 22.38  |    8.63     |   17.71    |     2.71      |   5.93    |    0     | 14.72 |
> | MInference         |  31.04  |   22   | 63.76  |  14.5   |  28.7  |    5.33     | **27.43** |     56.78     |   77.12   |    14    | 34.07 |
> | **Star Attention** | **31.85** | **25.92** | **69** | **22** | **30.37** | **24.37** |   26.29    | **93.22** | **96.27** | **45.8** | **46.51** |
>
> We thank the reviewer again for raising important points. Their comments led us to incorporate broader baselines and more diverse evaluations, which we believe have significantly strengthened the work. We look forward to incorporating these additions more prominently in the camera-ready version.

---

### Official Review · Reviewer_x5MA · 2025-03-14

**Overall Recommendation:** 1

**Summary:**

This paper presents Star Attention, a novel two - phase block - sparse approximation algorithm for efficient LLM inference over long sequences. The self - attention mechanism in Transformer - based LLMs has quadratic complexity, making long - sequence inference costly and slow. Star Attention addresses this issue by dividing the inference process into two phases. In Phase 1, the context is partitioned into blocks and distributed across multiple hosts. Each host computes local attention within its assigned block, which reduces the attention complexity from quadratic to linear. In Phase 2, the query is broadcast to all hosts, and global attention is computed at a designated query - host by aggregating local attention results. This approach enables the context length to scale linearly with the number of hosts. Experiments on Llama - based models show that Star Attention can achieve up to 11x faster inference speed compared to Ring Attention while maintaining 95 - 100% of the accuracy.

**Claims And Evidence:**

yes

**Essential References Not Discussed:**

no

**Experimental Designs Or Analyses:**

yes

**Methods And Evaluation Criteria:**

yes

**Other Comments Or Suggestions:**

no

**Other Strengths And Weaknesses:**

Strengths
1. Star Attention significantly speeds up LLM inference on long sequences. It manages to achieve a remarkable speedup, up to 11 times faster than the baseline in some cases. This is a huge improvement, especially considering the increasing demand for processing long - context data in applications like large - scale document analysis and multi - document summarization. For example, in the experiments with Llama - based models, it clearly outperforms the Ring Attention baseline in terms of inference time.
2. Despite the significant speed improvement, Star Attention can preserve 95 - 100% of the accuracy of global attention. This means that it doesn't sacrifice much in terms of the model's ability to understand and process the input accurately. Whether it's in simple retrieval tasks or more complex question - answering tasks, it can still provide reliable results.
3. The two - phase design is really smart. By separating context encoding and query encoding, it takes advantage of the characteristics of different parts of the input sequence. The use of anchor blocks in Phase 1 helps to manage the attention spikes and approximate global attention, which is a great way to optimize the attention mechanism. Also, the distributed softmax algorithm in Phase 2 enables efficient global attention computation without excessive communication overhead.
4. It is compatible with most Transformer - based LLMs trained with global attention. This means it can be easily integrated into existing models without the need for complex fine - tuning. This makes it very practical and convenient for researchers and engineers who want to improve the performance of their LLM - based systems.

Weaknesses
1. Although anchor blocks play a crucial role in Star Attention, there are still some aspects that need further exploration. For instance, the exact reason why the anchor block size needs to be equal to the context block size for optimal performance is not fully understood. Also, the relationship between the position and content of the anchor block and the model's performance could be studied more deeply. This lack of understanding may limit the further optimization of the algorithm.
2. In more complex tasks like Multi - Hop Tracing and some types of Question Answering tasks, Star Attention shows a slight decline in performance. These tasks require the model to have a deeper understanding of the context and often need inter - block communication. Since Star Attention lacks effective inter - block communication during context encoding, it struggles to perform as well as in simpler tasks. This means that there are still limitations when applying Star Attention to tasks that demand high - level context comprehension.
3. The performance of Star Attention is highly dependent on the block size. While setting the block size to one - quarter of the total sequence length seems to work well in most cases, using smaller blocks on longer sequences leads to accuracy degradation. This restricts the flexibility of the algorithm in different scenarios. For example, in some real - world applications where the sequence length may vary unpredictably, it may be difficult to choose the optimal block size.

**Questions For Authors:**

see weaknesses

**Relation To Broader Scientific Literature:**

no

**Theoretical Claims:**

yes

---

> ### Author Rebuttal · Authors · 2025-03-31
>
> We appreciate the reviewer x5MA's constructive feedback. Below, we address the identified weaknesses regarding anchor blocks, performance on complex tasks, and block size dependency:
>
> ### **1. Role and Configuration of Anchor Blocks:**
> - **Anchor Block Size:** The performance peak when the anchor block size equals the context block size can be attributed to Star Attention’s design constraints. Since Phase 1 uses causal attention (as in decoder-only LLMs), anchor blocks cannot extend beyond the preceding context block without violating causality. Setting the anchor size equal to the context block size represents the maximum usable context while respecting this constraint — effectively bringing Star Attention closer to full attention in terms of receptive field.
> - **Anchor Block Content and Position:** As shown in Section 4.1, using the first context block (`c1`) as the anchor yields optimal performance. The rationale is that Phase 2 performs global attention; therefore, Phase 1 context blocks should attend to anchor tokens representative of the initial sequence context that the full model would observe in a global attention scenario. Our analysis in Section 4.1 further indicates that if the anchor block content is fixed to `c1`, variations in the position IDs assigned to these anchor tokens during local attention computation have a minimal impact on overall performance.
>
> ### **2. Performance on Complex Tasks:**
> - Star Attention, like other sparse attention mechanisms, trades off some accuracy for substantial inference speedups. Across a range of tasks — including question answering (Figure 7) — it retains 95–100% of dense attention performance while delivering substantial inference speedups.
> - For tasks requiring complex reasoning and inter-block context (e.g., Multi-Hop Tracing), we acknowledge that the absence of direct cross-block attention in Phase 1 introduces challenges. Despite this, Star Attention maintains up to 93% of the dense attention accuracy on these tasks (Figure 7).
> - To assess generalization on more complex, real-world tasks, we evaluated Star Attention on InfiniteBench, a diverse benchmark covering multilingual QA, retrieval, math, code debugging, summarization and more. As shown in Table B, Star Attention matches or closely tracks full attention across all categories and outperforms prior sparse inference baselines by a significant margin.
>
> **Table B: Accuracy on Infinite Bench (Llama-3.1-8B-Instruct)**
> | Methods            | En. Sum | En. QA | En. MC | En. Dia | Zh. QA | Code. Debug | Math. Find | Retr. PassKey | Retr. Num | Retr. KV |  Avg. |
> | :----------------- | :-----: | :----: | :----: | :-----: | :----: | :---------: | :--------: | :-----------: | :-------: | :------: | :---: |
> | *Full Attn.  (Baseline)* | *31.91* | *25.92* | *69.43* | *21.5* | *31.95* | *16.75* | *24.29* | *99.15* | *99.66* | *60* | *48.06* |
> | StreamingLLM       |  30.15  | 10.15  | 41.05  |   8.5   | 22.38  |    8.63     |   17.71    |     2.71      |   5.93    |    0     | 14.72 |
> | MInference         |  31.04  |   22   | 63.76  |  14.5   |  28.7  |    5.33     | **27.43** |     56.78     |   77.12   |    14    | 34.07 |
> | **Star Attention** | **31.85** | **25.92** | **69** | **22** | **30.37** | **24.37** |   26.29    | **93.22** | **96.27** | **45.8** | **46.51** |
>
> - Improving the performance on tasks requiring deeper inter-block reasoning in the early encoding stages remains an important direction for future work.
>
> ### **3. Dependency on Block Size and Flexibility:**
> - The dependency on block size reflects a common trade-off in blockwise attention: smaller blocks enable faster inference but may limit context aggregation, especially in longer sequences. Our experiments (Figure 5) show that setting the block size to ~1/4 of the total sequence length achieves a strong balance of accuracy and speed. However, the block size depends on the user and they can choose smaller block sizes to prioritize higher inference speeds, accepting slightly more accuracy degradation.
> - For variable-length or real-time inputs, users can set the block size based on the maximum expected sequence length or system constraints. As shown in Figure 6 and Table 5, even with smaller blocks (1/8, 1/16, 1/32 of the sequence), Star Attention retains up to 90% of dense attention performance while delivering up to 17× speedups — allowing users to flexibly tune accuracy vs. latency based on application needs.

---

### Decision · Program_Chairs · 2025-05-01

**Decision:**

Accept (poster)

**Comment:**

The paper introduces a practical and efficient method for long-sequence LLM inference, offering substantial speedups (up to 11×) while maintaining high accuracy (95–100%). It is compatible with existing Transformer models and demonstrates strong performance compared to Ring Attention and Sparse Attention. The two-phase design is well-motivated, and the empirical results—strengthened by a thorough rebuttal—show consistent performance across diverse tasks and benchmarks. However, concerns remain regarding the method's novelty, with some reviewers noting similarities to existing sparse attention mechanisms.

The submission has polarized reviewers. One reviewer recommends acceptance, highlighting originality and empirical strength; another leans toward rejection due to perceived lack of novelty and insufficient motivation. Two reviewers rate the paper as weak rejects but appreciate its engineering contributions.

Despite some valid critiques, the paper demonstrates clear technical contributions, practical relevance, and thorough empirical validation. The authors actively addressed all reviewer concerns during rebuttal, and the method’s scalability and compatibility with existing LLM architectures are compelling.

The paper should be accepted, conditional on improving discussion around prior work (especially streaming variants and attention sparsification techniques), and clarifying performance metrics (e.g., decoupling memory vs. speedup claims). A more comprehensive system-level analysis could further strengthen the paper.